# An Operation Guide of Si-MOS Quantum Dots for Spin Qubits

**DOI:** 10.3390/nano11102486

**Published:** 2021-09-24

**Authors:** Rui-Zi Hu, Rong-Long Ma, Ming Ni, Xin Zhang, Yuan Zhou, Ke Wang, Gang Luo, Gang Cao, Zhen-Zhen Kong, Gui-Lei Wang, Hai-Ou Li, Guo-Ping Guo

**Affiliations:** 1CAS Key Laboratory of Quantum Information, University of Science and Technology of China, Hefei 230026, China; hrz@mail.ustc.edu.cn (R.-Z.H.); rlma@mail.ustc.edu.cn (R.-L.M.); mingni@mail.ustc.edu.cn (M.N.); xzhang16@mail.ustc.edu.cn (X.Z.); zy1995@mail.ustc.edu.cn (Y.Z.); wk0910@ustc.edu.cn (K.W.); rogone@ustc.edu.cn (G.L.); gcao@ustc.edu.cn (G.C.); gpguo@ustc.edu.cn (G.-P.G.); 2CAS Center for Excellence and Synergetic Innovation Center in Quantum Information and Quantum Physics, University of Science and Technology of China, Hefei 230026, China; 3Key Laboratory of Microelectronics Devices & Integrated Technology, Institute of Microelectronics, Chinese Academy of Sciences, Beijing 100029, China; kongzhenzhen@ime.ac.cn; 4Origin Quantum Computing Company Limited, Hefei 230026, China

**Keywords:** Si-MOS, quantum dot, spin qubits, quantum computing

## Abstract

In the last 20 years, silicon quantum dots have received considerable attention from academic and industrial communities for research on readout, manipulation, storage, near-neighbor and long-range coupling of spin qubits. In this paper, we introduce how to realize a single spin qubit from Si-MOS quantum dots. First, we introduce the structure of a typical Si-MOS quantum dot and the experimental setup. Then, we show the basic properties of the quantum dot, including charge stability diagram, orbital state, valley state, lever arm, electron temperature, tunneling rate and spin lifetime. After that, we introduce the two most commonly used methods for spin-to-charge conversion, i.e., Elzerman readout and Pauli spin blockade readout. Finally, we discuss the details of how to find the resonance frequency of spin qubits and show the result of coherent manipulation, i.e., Rabi oscillation. The above processes constitute an operation guide for helping the followers enter the field of spin qubits in Si-MOS quantum dots.

## 1. Introduction

As early as 1982, the famous physicist Feynman proposed that quantum computers can simulate problems that cannot be solved by classical computers [1]. Then, in 1994, Shor proposed the well-known quantum prime factor decomposition algorithm that can be used to crack classic RSA encrypted communications [2], and in 1996, Grover devised the quantum search algorithm which uses only $O(\sqrt{N})$ evaluations of the function [3]. After that, Loss and DiVincenzo proposed the Loss–DiVincenzo quantum computer in 1998 [4] and then in 2000, DiVincenzo presented the DiVincenzo Criteria for physical implementation of quantum computing [5]. These findings set off a wave of quantum computing research.

In this wave, researchers tried to build quantum computers in various systems. Trapped ions [6,7], nuclear magnetic resonance (NMR) [8,9], superconducting loops [10,11], nitrogen vacancy center [12,13], semiconductor quantum dots [14,15], and other systems have enabled the manipulation of single and two qubits and have demonstrated simple quantum algorithms. Among them, silicon quantum dots (QDs) have emerged as promising hosts for qubits to build a quantum processor due to their long coherence time [16,17], small footprint [18], potential scalability [19,20], and compatibility with advanced semiconductor manufacturing technology [21].

In recent decades, silicon QDs have engaged research participants all around the world and have developed fast. In 2012, a long-time singlet–triplet oscillation was realized in silicon double quantum dots (DQD) [22]. Then, high-quality single-spin control was developed in silicon QDs [16,23]. After that, a two-qubit controlled gate in silicon QDs was experimentally implemented [24,25,26,27]. Nowadays, the single-qubit operations of spin qubits achieve fidelities of 99.9% [28,29], the two-qubit operation fidelities are above 99% as reported [30], the spin–photon coupling rates are more than 10 megahertz [31,32,33], and the qubit operation temperature is higher than 1 kelvin [34,35]. In the meantime, experiments on other properties of silicon QDs, including valley states [36,37,38], orbital states [39], and noise spectra [40,41], have been carried out. Furthermore, experimental approaches and techniques for characterizing features of QDs from other systems, e.g., charge stability diagrams [42,43,44], random telegraph signals (RTS) [45,46,47,48], Elzerman readout [49,50], Pauli spin blockade (PSB) readout [51,52], electron spin resonance (ESR) [53,54], and electron dipole spin resonance (EDSR) [55,56], have been applied in silicon QDs as well. In addition, several reviews [57,58,59] and guides on fabrication [60] have been reported. However, the process from silicon QD to qubit manipulation is still challenging.

In this article, we give a brief introduction of how to realize a single spin qubit from QDs in a Si-MOS structure. First, we introduce the gate-defined DQD in an isotopically enriched ${}^{28}$Si-MOS structure and the low-temperature measurement circuits. Second, by applying these circuits, we investigate the basic properties of silicon QD devices, i.e., charge states, excited orbital states, valley splitting, lever arms, electron temperature, tunneling rate, and noise spectrum. Then, we introduce two mainstream spin-state-readout methods named as the Elzerman readout and the PSB readout. Finally, we use the rapid adiabatic passage to find out the resonance frequency of the spin qubits and apply the Rabi pulsing schemes to coherently manipulate the spin qubit.

## 2. Materials and Methods

### 2.1. Spin Qubit Devices

Spin qubits are hosted in a pair of metal-oxide-semiconductor (MOS) dots with isotopically enriched silicon. By using the high vacuum activation annealing technique, we improve the mobility of Si-MOS devices by a factor of two, reaching 1.5 $m^{2}$/(V·s) [61]. In this work, we use a DQD that has a similar structure (Ref. [38]) and was fabricated in our lab’s clean room. As shown in Figure 1a, the aluminum electrodes are vaporized on top of the silicon oxide by electron beam evaporation techniques. Between every two layers of the electrodes, an insulating layer of aluminum oxide is formed by thermal oxidation. Figure 1b shows that the electrons are confined in the potential wells and the DQD is formed by applying voltages to the electrodes [62]. In the quantum well, a single electron can tunnel between the two QDs by biasing the electrodes’ voltages. The entire structure consists of a DQD and a single-electron transistor (SET) sensing the charge states in DQD.

### 2.2. Measurement Circuits

There are three main types of measurement circuits commonly used to characterize the properties of DQDs, as shown in Figure 2a,c,e:Figure 2a: Transport measurements based on a lock-in amplifier. The AC excitation is added to the SET source ($S_{1}$) by connecting the lock-in amplifier to an external 1000:1 voltage divider, and finally reaches $S_{1}$ at approximately 50 μV, with a lock-in frequency generally between 70 and 1000 Hz. In addition, the drain ($D_{1}$) is connected back to the lock-in amplifier to demodulate the signal and obtain the currents.Figure 2c: Charge detection based on the lock-in amplifier. The bias voltage at $S_{1}$ is connected to a Stanford Research Systems Isolated Voltage Source (SIM 928) through a 1000:1 voltage divider, reaching $S_{1}$ at around 500 μV, while the AC excitation of the lock-in amplifier (output at approximately 0.5 to 1.5 mV) is connected to LP through an analog summing amplifier (SIM 980, bandwidth of approximately 1 MHz).Figure 2e: Charge detection based on a current-voltage amplifier. The source-drain bias is the same as Figure 2c, except no excitation is applied to the LP and $D_{1}$ is connected to a current-voltage amplifier; here, we use a Femto DLPCA200, connected to a voltage amplifier (SIM 910), an analog filter (SIM 965), and finally to a voltmeter (Agilent 34410) for signal measurement or to a PCI-based waveform digitizer (ATS 460), oscilloscope, etc. for the real-time observation of electron tunneling.

## 3. Results

### 3.1. Basic Properties of Silicon QDs

#### 3.1.1. Charge Stability Diagram

Obtaining the QD charge stability diagram by the charge detection method is one of the most basic QD characterization methods [42,43,44]. As shown in Figure 2a–d, according to the method of measuring QDs using the modulation signal of the lock-in amplifier introduced in Section 2, the source ($S_{2}$) and drain ($D_{2}$) of the DQD are grounded. We set the voltages of the SLB and SRB near a Coulomb peak so that the SET works sensitively at this position, which is identified by the yellow star in Figure 2b. Then, a voltage of 2.60 V is applied to the LL and RL to ensure that the channel of DQD is turned on. After that, we measured the charge stability diagrams with different gate voltages to obtain the DQD electron occupation numbers and tunneling properties of the left QD. Figure 2d shows the charge stability diagram of the last few electrons in the DQD. The numbers in this figure indicate the electron occupation on the left and right QD. The slope is relatively symmetric with respect to the two electrodes. This indicates that two QDs are formed under electrodes LP and RP. When scanning the voltage of LB and LP, there are continuous electron tunneling lines observed, which correspond to the left QD. As shown in Figure 2f, the tunneling line of the last few electrons in the left QD becomes more invisible when the voltage of LB decreases. This is because a decrease in the LB voltage reduces the tunneling rate of the left QD to the reservoir of $D_{2}$.

#### 3.1.2. Detection of Orbital Excited States in Silicon QDs

The orbital excited state in silicon QDs is several meV above the ground state, and it can be detected by the pulsed-voltage spectroscopy method [39,63]. Based on the measurement circuit in Figure 2c, we change the modulation signal output from the lock-in amplifier to a square waveform generated by an external arbitrary waveform generator that is synchronized with the lock-in amplifier. By zooming in and remapping, the single tunneling lines in Figure 3a split into pairs of lines in Figure 3b. As shown in Figure 3c, the principle of the pulsed-voltage spectroscopy method is illustrated. When the voltage of LP is set at the position of the blue square in Figure 3d, the electron can tunnel into the ground state of the QD. As the voltage increases, the energy level of the excited state gradually approaches the amplitude window of the square wave. When the excited state enters the window, the electron can tunnel into the excited state, so that another transport line appears parallel to the left line, identified by the green circle in Figure 3d. According to Figure 3d and the extracted lever arm of LP ($\alpha_{LP}$, which will be discussed in Section 3.1.3), which is 0.33 meV/mV, the calculated energy of excited state is 1.3 meV.

#### 3.1.3. Detection of Valley States in Silicon QDs

In solid-state physics, due to the six-fold degeneracy at the bottom of the conduction band of silicon, the energy levels at the bottom of the six conduction bands are named as the valley level. In the case of two-dimensional electron gas, the six-fold degeneracy is split into a four-fold degeneration and a double-fold degeneration. Due to the existence of the interfacial electric field, the quadruple degenerate and the double degenerate split further and form valley-level splits [57]. Unlike the orbital state, the splitting energy of the two lowest valley states ($E_{VS}$) in silicon QDs is similar to the Zeemen splitting energy ($E_{Z}$) under the applied magnetic field in our experiment [36,37,38,39,64,65]. Therefore, it is important to determine the splitting energy of the valley state. A commonly used method is to measure the electron tunneling line at different magnetic fields. Here, we tune the energy level of the first four electrons by changing the magnetic field of which the direction of is along the surface of the device and perpendicular to the one-dimensional channel formed by the QD, as shown in Figure 1a. Figure 4a shows the transition lines of the first four electrons in the device in Ref. [38]. The voltage of the first transition line of the QD decreases as the magnetic field increases, while the fourth line increases. Differently, the second transition line increases first and then decreases, and the third line is reversed to the second line.

We use the principle of minimum energy to simply explain this phenomenon, as shown in Figure 4b. When filling the first electron, the electron will be filled to the lowest energy level. As the magnetic field increases, $E_{Z}$ increases, so the energy level of filling the first electron decreases. When filling the second electron, the first electron has been filled to the bottom level. In accordance with the principle of minimum energy, the second electron should be filled with the second-lowest level, but this second-lowest energy level depends on the magnetic field. When the magnetic field is small, the second-lowest energy state is the valley state $v_{-}$ with spin state up, and vice versa. As shown in Figure 4b, the energy states of the third and fourth electron are mirror-symmetrical to the second and first electrons, respectively. It is obvious from Figure 4b, the position of the kink point is exactly where $E_{VS}$ is equal to $E_{Z}$, so by using the position of the kink point and the Bohr magneton ($\mu_{B}$), we can obtain:(1)$$E_{VS}=g\mu_{B}B_{\mathrm{kink}}$$

According to Figure 4a, the $E_{VS}$ of the second electron is 170 μeV, and the $E_{VS}$ of the third one is 245 μeV. The difference between the $E_{VS}$ of these two electrons is caused by the different LP voltages [38].

Additionally, we can estimate $\alpha_{LP}$ by:(2)$$\alpha_{LP}=\frac{g\mu_{B}\Delta B}{2\Delta V_{LP}}$$

Therefore, the lever arms of the first four electrons are shown in Table 1:

### 3.2. Real-Time Observation of Electron Tunneling in Silicon QDs

The characterization of the orbital state, spin state, and valley state in QDs is based on steady-state measurement. However, to detect more properties of electrons, such as tunneling rate, electron temperature, noise spectrum, spin state, etc., we also need the ability to observe the tunneling process of electrons in QDs in real time [45,46,47,48]. The measurement circuit of real-time detection has been introduced in Section 2, as shown in Figure 2e. Next, we introduce the measurement results of tunneling rate, electron temperature, noise spectrum, and spin state, respectively.

#### 3.2.1. RTS and the Measurement of Electron Temperature

When we align the electrochemical potential of the first electron in the QD with the Fermi surface of the electron reservoir, the electrons will continuously tunnel in and out of the QD (see the green circle in Figure 5a,b). At this time, on the oscilloscope or digitizer, we can see the signal as shown in the inset of Figure 5b. Since tunneling events happen randomly, we call the observed signal a RTS.

Ideally, electrons tunnel only when the electrochemical potential in the QD is aligned with the Fermi surface of the electron reservoir. However, in practice, due to the limited electron temperature, the Fermi surface of the electron reservoir will have a certain broadening. Therefore, changes in the electron tunneling events can be observed when the LP voltage is changed. The insets in Figure 5b show that when the electrode voltage increases, the electrochemical potential in the QD decreases, so the probability of the electrons occupying the energy state in the QDs gradually increases and vice versa. By fitting the Fermi distribution to the electron occupancy, we can extract the electron temperature. The specific form of the Fermi distribution function we used here is the following [48]:(3)$$N=\frac{1}{\exp\left[\alpha_{LP}\left(V_{LP0}-V_{LP}\right)/\left(k_{B}T\right)+1\right]}$$
where $k_{B}$ represents the Boltzmann constant, $\alpha_{LP}$ has been calculated in Table 1, $V_{LP0}$ and *T* are fitting parameters. By fitting this equation, the electron temperature of approximately 224 mK is obtained.

#### 3.2.2. Measurement of the Tunneling Rate

For the RTS, we can mark the time of electron tunneling from the reservoir to the QDs as $t_{on}$, and the time of electron tunneling from the QDs to the reservoir as $t_{off}$. By counting the distribution of $t_{on}$ and $t_{off}$ over a long period of time, we can actually determine the time of electron tunneling in and out of the reservoir [45].

Here, we introduce another method based on RTS. As shown in Figure 5c, by applying a square waveform on the LP, the signal will also switch between low and high levels with an approximate square wave period. Figure 5d illustrates that the rising and falling edges of the signal are slower, unlike the square wave from the AWG. Excluding the bandwidth limitation of the SET, the width of the edges represents the electron tunneling times $t_{on}$ and $t_{off}$. By fitting the rising and falling edges exponentially, we can obtain the exact tunneling time values: $t_{on}=3.45$ ms and $t_{off}=3.23$ ms.

#### 3.2.3. Noise Spectrum

When observing the real-time electron tunneling signal, there will inevitably be noise interference. Analyzing the noise spectrum can help us analyze the source of the noise and then suppress the noise. Figure 6a shows a typical noise spectrum of a QD system but does not include the noise introduced by the measurement system. The QD system suffers from charge noise [29], random telegraph noise (RTN) [40] and nuclear noise [66] at low frequencies. Johnson Nyquist noise and phonon noise are relatively large at high frequencies and affect the spin relaxation time.

Figure 6b shows the noise spectrum under the different device conditions given in Ref. [38]. The red line is the noise spectrum when the QD is connected. This noise conforms to the law of 1/f. In fact, this is typical charge noise from the QD. The green line is the noise spectrum when the QD is not connected and almost overlaps with the red line above 10 Hz, indicating that the noise above 10 Hz does not come from the QD. The blue line is the noise spectrum when the amplifier input is open, indicating that the noise above 10 Hz comes from the DC line. Compared with Ref. [41], the noise from the QD is low enough for the measurement of the qubit. On the other hand, the capacitance and resistance of the DC line is the main reason for this high-frequency noise. To further reduce the noise source, we can switch to a coaxial line with a smaller capacitance in the future.

### 3.3. Spin State Readout

After being able to detect the QD charge state and control and measure the tunneling of electrons through a simple square waveform, we now introduce two of the most commonly used methods for spin-to-charge conversion: the Elzerman readout [49,50] and PSB readout [14,51,52,53].

#### 3.3.1. Elzerman Readout

The process of the Elzerman readout is shown in Figure 7a. We set the voltage to locate the Fermi surface of the electron reservoir between the energy state of the electrons with different spin states. Therefore, the spin-up electrons can tunnel to the electron reservoir (after a period of time to load spin down electrons from the electron reservoir), while the spin down electrons cannot. Since the signal of SET responds to the two events of electron tunneling in and out of the QD, a square wave is formed in the signal of the SET. By observing the change in the current, it can be determined whether electron tunneling occurs; then, it can be determined whether the spin state of the electron is up.

Figure 7b shows a series of the measured SET current signal while reading the spin state. The signal in the top panel has a square pulse, which corresponds to a spin up state. The signal in the bottom panel does not have such a pulse and indicates a spin down state. Based on the above process, we have achieved a single-shot readout of the electron spin state.

#### 3.3.2. PSB

For the PSB readout, we first need to know the double spin eigenstates; the singlet (S) and triplet (T, include $T_{0}$, $T_{+}$ and $T_{-}$) states:(4)$$S=\frac{|↑↓\rangle -|↓↑\rangle }{\sqrt{2}},T_{0}=\frac{|↑↓\rangle +|↓↑\rangle }{\sqrt{2}},T_{+}=|↑↑\rangle ,T_{-}=|↓↓\rangle$$

When there is no magnetic field, the singlet state is the ground state. The three-spin triplet state energies degenerate, which is referred to as the T state. This T state is an excited state. Now, we consider two charge states in a DQD: (1,1) and (0,2). For the (0,2) state, there are two electrons in one QD. According to the Pauli exclusion principle, the spin wave function of the electrons in the T state is symmetric, so two electrons must occupy different orbital states. Therefore, S(0,2) and T(0,2) are non-degenerate, as shown in Figure 7c,d. $\Delta_{ST}$ is the energy difference between S(0,2) and T(0,2). However, for the (1,1) state, two electrons are located in their respective QDs, thus avoiding the Pauli exclusion principle and two electrons can occupy one orbital state. Therefore, S(1,1) and T(1,1) are almost degenerate, as shown in Figure 7c,d.

Based on these energy states, we now introduce the PSB readout. As shown in Figure 7c, when a negative bias is applied (the Fermi surface of the source is higher than the drain), electrons in the source can first tunnel to the S(1,1) or T(1,1) state. When tunneling to the S(1,1) state, the electron can continue to tunnel to S(0,2) and then reach the drain to form current. When tunneling to the T(1,1) state, the electron cannot continue to tunnel to S(0,2) due to the PSB, and T(0,2) is higher than T(1,1), so the electron cannot enter any (0,2) charge states, and the current is suppressed. Figure 7d shows the positive bias condition. The Fermi surface of the source is lower than the drain. The electrons in the drain tunnel to the S(0,2) state, and then through the S(1,1) state to the source to form a current. No blockade occurs in the process, so there is current in the entire bias triangle region. In addition, for the PSB readout, we can use the SET to sense the charge states in DQD, and the operating temperature can be raised to higher than 1 kelvin [34,35].

#### 3.3.3. Measurement of Spin Lifetime

After being able to perform a single-shot measurement to read the spin state, we can use the same waveform to measure the spin lifetime ($T_{1}$) [50]. The process of a typical single-shot readout is shown in Figure 8a. First, we reduce the voltage for electron evacuating from the QD; this is also referred to as “empty”. Then, we raise the voltage so that electrons can tunnel from the electron reservoir to the QDs, which is also called “load”. At this time the spin state of the electron in QD is random. Finally, we carefully reduce the voltage to locate the Fermi surface of the electron reservoir between the energy state of different spin electrons to “read” the spin state. We count the number of spin relaxation events for different load time periods. Figure 8b illustrates that the probability of the spin up state ($P_{↑}$) decreases exponentially, so the $T_{1}$ can be obtained by fitting the exponential function.

### 3.4. Manipulation of the Spin Qubit

Now that we are able to read the spin state via the single-shot readout method, we introduce the manipulation of the spin qubit. There are two mainstream manipulation methods: ESR [16,53,54] and EDSR [23,52,55,56,67,68]. The ESR can be achieved by applying an alternating magnetic field $B_{1}$(5–50 μT) perpendicular to the external magnetic field $B_{\mathrm{ext}}$ (typically 150–1500 mT) via an antenna structure. For EDSR, we apply an alternating electric field combined with spin-orbit coupling to flip the spin. However, the natural spin–orbit coupling in silicon is weak, so we need micromagnets to introduce a gradient magnetic field to construct synthetic spin-orbit coupling. The advantages of EDSR include a fast spin flip rate, low heating, ease of fabrication, etc. However, the additional magnetic field from the micromagnets makes it difficult to find the resonance frequency $\gamma_{e}\left(B_{\mathrm{ext}}+B_{1}\right)$ of the qubit. Therefore, we introduce rapid adiabatic passage to solve this problem.

#### 3.4.1. Rapid Adiabatic Passage

We use frequency chirped microwave bursts, and when the excitation frequency passes through the resonance frequency, the electron spin is inverted (see Figure 9b) [23,55,56]. Figure 9a shows the principle of the rapid adiabatic passage process. In the reference frame rotating at the resonance frequency, the Hamiltonian of the system is the following [56]:(5)$$H\left(t\right)=\frac{1}{2}\frac{\partial }{\partial t}\left(\Delta \nu \right)t\sigma_{z}+\nu_{1}\sigma_{x}$$

Here, Δν is the microwave frequency detuning from the resonance frequency, and $\nu_{1}$ is the spin flip rate.

We use the Landau–Zener theory to solve this time evolution of a two-level system that is described by a linearly time-dependent Hamiltonian. The probability of adiabatic transition from one eigenstate to the other is given by [56]
(6)$$P=1-\exp\left(-4\pi^{2}\frac{\nu_{1}^{2}}{\left|\frac{\partial }{\partial t}\left(\Delta \nu \right)\right|}\right)$$

An electron spin in the |↓〉 state will flip to the |↑〉 state if the microwave frequency sweeps across the resonance frequency. To satisfy the adiabatic evolution condition, the sweep rate $\left|\frac{\partial }{\partial t}\left(\Delta \nu \right)\right|$ cannot be too fast compared with $\nu_{1}$.

#### 3.4.2. Rabi Oscillation

After calibrating the resonant frequency through the rapid adiabatic passage, we now use a single-frequency microwave combined with a single-shot readout to manipulate the qubit [54,56], as shown in the inset of Figure 10b. Figure 10a shows the Rabi pulsing scheme. First, we increase the voltage so that electrons in the |↓〉 or |↑〉 state cannot tunnel from the QDs to the electron reservoir. We apply the microwave pulse before the next stage to flip the electron spin. Then, as mentioned in Section 3.3.3, we carefully decrease the voltage to locate the Fermi surface of the electron reservoir between the energy states of different spin electrons to “read” the spin state. At the end of the “read” phase, the electron spin state will be |↓〉 no matter the spin state at the beginning. Figure 10b shows the result of a Rabi oscillation. As the microwave duration time increases, the spin of the qubit continuously flips between |↓〉 and |↑〉 states. The amplitude of oscillation decreases with time due to noise. We fit the Rabi oscillation with the function $P\left(t\right)=A\cdot \exp\left(-t/T_{2}^{\mathrm{Rabi}}\right)\cdot \sin\left(f_{\mathrm{Rabi}}t\right)$. Here, $f_{\mathrm{Rabi}}=1.256\pm 0.003$ MHz represents the spin flip rate, and $T_{2}^{\mathrm{Rabi}}=5.4\pm 0.4\;\mu$s represents the influence of the noise in Figure 10b.

## 4. Conclusions

In this paper, we provide an operation guide of Si-MOS QDs for spin qubits. First, we introduce the structure of the devices and the measurement circuit. Next, we show the charge stability diagram and detect the orbital and valley states. Then, we use a digitizer to detect the RTS and measure electron temperature and tunneling rate. Moreover, we introduce two commonly used methods, the Elzerman readout and the PSB readout, and use the single-shot readout method to measure the $T_{1}$. Finally, we give a brief introduction of ESR and EDSR, use rapid adiabatic passage to calibrate the resonance frequency of the spin qubit, and show the result of the Rabi oscillation. For future directions, researchers may be interested in hybrid qubits coupling [33], hot qubits [34,35], cryogenic control [69,70], foundry-fabrication [71,72], high fidelity readouts [73,74], and qubit number expansion [75].

## Figures and Tables

**Figure 1 nanomaterials-11-02486-f001:**
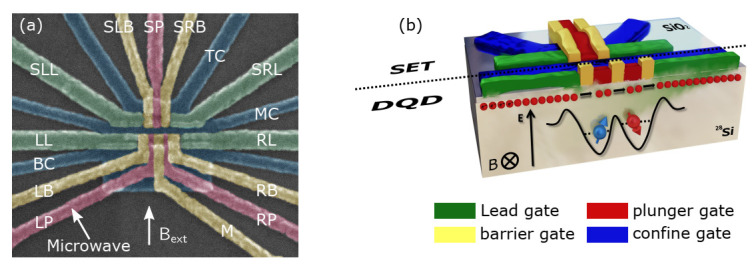
(**a**) Scanning electron micrograph image of a typical Si-MOS DQD. An SET, which is used as a charge sensor, is confined by the top confine gate (TC), middle confine gate (MC), left barrier gate (SLB), and right barrier gate (SRB) and is tuned by the plunger gate (SP). A DQD is composed of a left lead gate (LL), right lead gate (RL), left barrier gate (LB), middle gate (M), right barrier gate (RB), left plunger gate (LP), and right plunger gate (RP) and is confined by a bottom confine gate (BC) and middle confine gate (MC). We tune the left and right QD via the LP and RP, respectively. The tunneling rate of the QDs can be tuned by the LB and RB. The spin of electrons in the left QD is controlled by applying a microwave pulse to the LP. The right white arrow indicates the direction of an in-plane external magnetic field. (**b**) Cross-sectional view of a 3D model of the device. Electrodes for different functions are distinguished by different colors. The SET and DQD are on each side of the dotted line. The electrons in the DQD are located under the plunger gates.

**Figure 2 nanomaterials-11-02486-f002:**
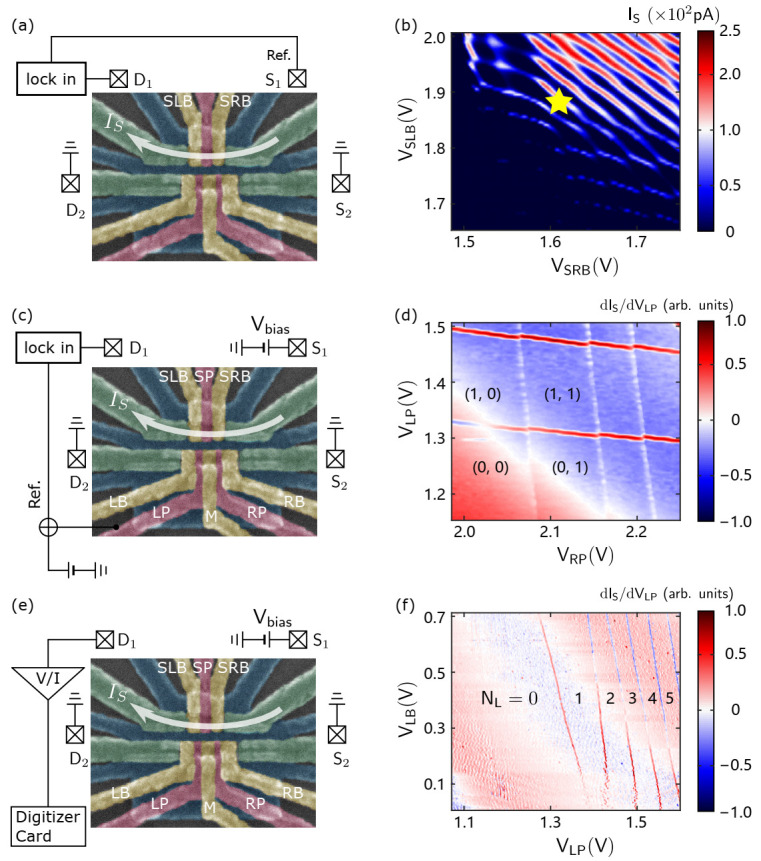
The three different measurement circuits. The white arrow above the SET indicates the direction of the current ($I_{S}$). (**a**,**b**) Measurement circuit diagram of the SET using the lock-in amplifier and the Coulomb peak diagram obtained by scanning the voltage of the SET barrier gates (SRB and SLB). The yellow star identifies a sensitive SET position. (**c**,**d**) Measurement circuit diagram of the DQD and the charge stability diagram of the DQD obtained by scanning the RP and LP. (**e**,**f**) Measurement circuit diagram for measuring the DQD using a current-voltage amplifier and the corresponding charge stability diagram of the left QD.

**Figure 3 nanomaterials-11-02486-f003:**
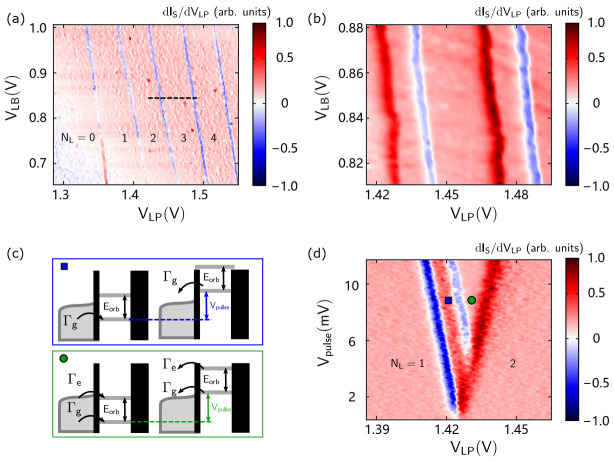
(**a**) Charge stability diagram of the QDs obtained by scanning the LB and LP voltages. (**b**) Zoom in on the QD charge stability diagram after applying a square pulse with a frequency of 687 Hz and an amplitude of 20 mV. (**c**) Schematic diagram of the square pulse spectrum measurement of the excited orbital state. When the LP voltage increases, the tunneling line of the electron in the excited state appears. (**d**) Diagram of the excited orbital state obtained by scanning the amplitude of the square pulse and the LP voltage.

**Figure 4 nanomaterials-11-02486-f004:**
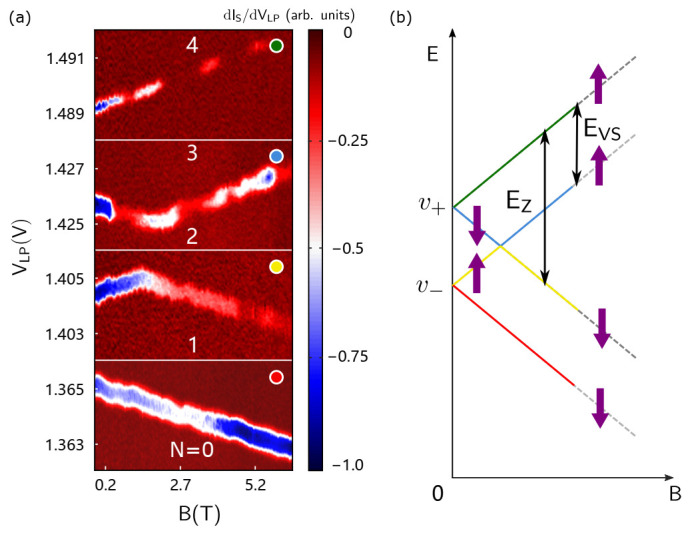
The magnetic spectrum and the corresponding diagram of the energy state for different electron numbers in the QDs. (**a**) The dependence of different electron tunneling lines on the magnetic field, where N = 0, 1, 2, 3, and 4 refer to the number of electrons in the QD. The slopes of the first four electron tunneling lines reveal the lever arms of LP for the first four electrons. (**b**) Energy state diagram of $E_{Z}$ as a function of the magnetic field. The ordinate is the state energy, and the abscissa is the order of the magnetic field. The purple arrow indicates the direction of the spin. The arrow with $E_{VS}$ represents the energy of the valley splitting.

**Figure 5 nanomaterials-11-02486-f005:**
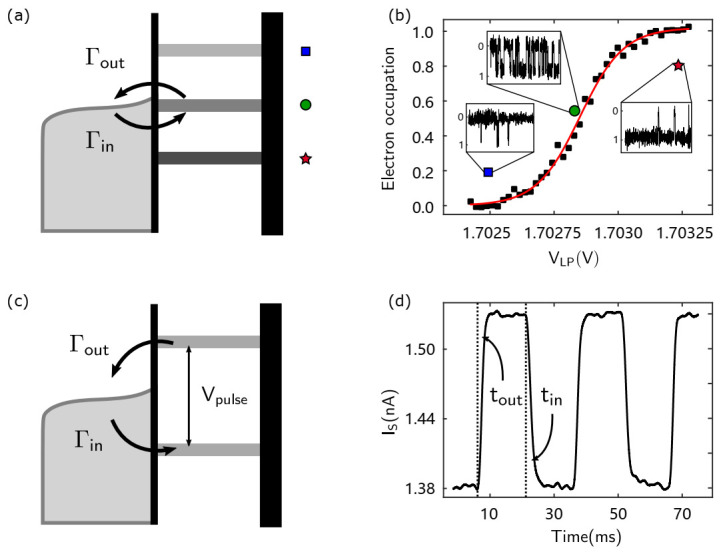
Measurement of the electron temperature and tunneling rate. (**a**) Schematic diagram of different positions of the QD energy state and the Fermi surface of the electron reservoir. (**b**) The probability of electron occupation probability. The electron temperature can be fitted as 223.8 ± 0.8 mK. The inset shows RTS for different electron occupation situations: the circle, square and star marks correspond to the alignment, negative bias and positive bias, respectively. (**c**) Electron tunnel in the QD when the voltage is high and vice versa. (**d**) The average current of the electron tunneling by applying a square wave with a 30 ms period. The average current decays exponentially with the tunnel time, and is characteristic of a Poisson process. A single exponential fitting can be used to obtain $t_{on}$ and $t_{off}$.

**Figure 6 nanomaterials-11-02486-f006:**
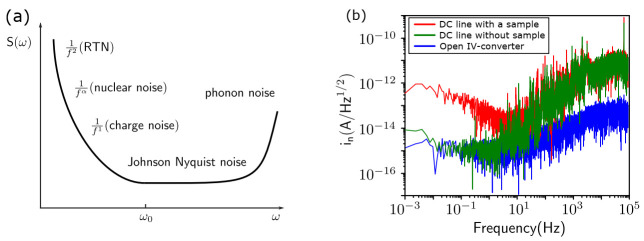
Measurement of the noise spectrum in silicon QD. (**a**) Typical noise spectrum of a silicon QD; the noise from the measurement system is not included. Here, $\omega_{0}$ is the spin resonance frequency. (**b**) The noise spectrum is measured by a dynamic signal analyzer (SR785) in our system and the spectrum contains three conditions: DC line with a sample, DC line without sample and open I–V converter.

**Figure 7 nanomaterials-11-02486-f007:**
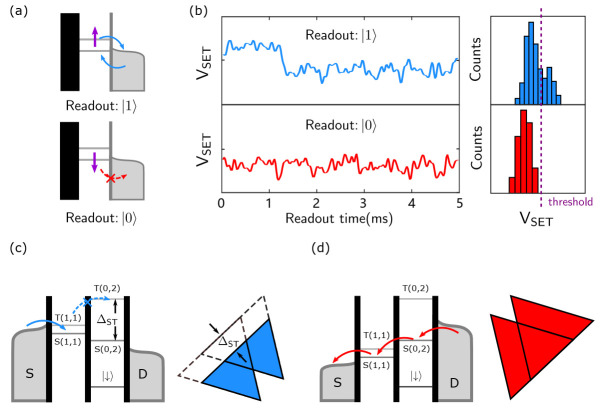
Spin-charge conversion. (**a**) Schematic diagram to read the spin state by the Elzerman readout. The spin-up electrons can tunnel out because the energy is higher than the Fermi surface of the electron reservoir and vice versa. (**b**) The measurement result of the spin state is read out by the Elzerman method in our experiment. When the electron in spin-up state tunnels out, there is a high level in the signal. The electron in spin-down state cannot tunnel out, so the signal remains at a low level. (**c**) Schematic diagram of the energy state and corresponding measurement results of the electron transition current with a negative bias. $\Delta_{ST}$ is the energy difference between the S and T states. When the energy detuning is less than $\Delta_{ST}$, PSB occurs. (**d**) Schematic diagram of the energy state with a positive bias. Here, no PSB occurs.

**Figure 8 nanomaterials-11-02486-f008:**
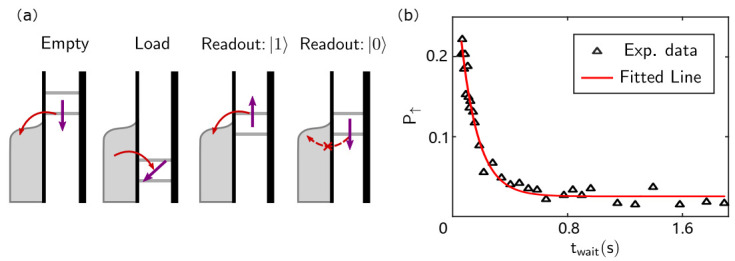
Schematic diagram and measurement result of $T_{1}$. (**a**) Schematic diagram of a single-shot readout for $T_{1}$ measurement. (**b**) Measured spin up probability ($P_{↑}$) as a function of waiting time ($t_{\mathrm{wait}}$). The fitting result of $T_{1}$ is 335±5 ms for the left QD.

**Figure 9 nanomaterials-11-02486-f009:**
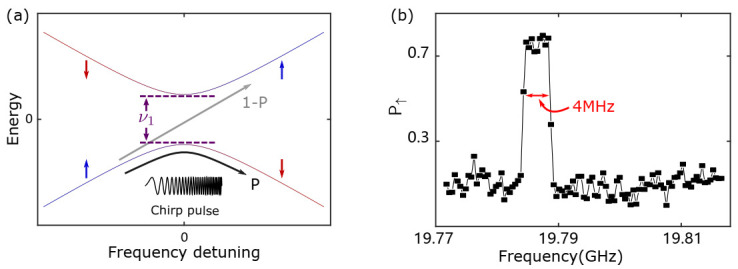
Schematic explanation and measurement result of rapid adiabatic passage. (**a**) Schematic explanation of rapid adiabatic passage in the rotating reference frame. (**b**) $P_{↑}$ as a function of microwave frequency with a 0.5 ms burst time and a 4 MHz frequency modulation depth.

**Figure 10 nanomaterials-11-02486-f010:**
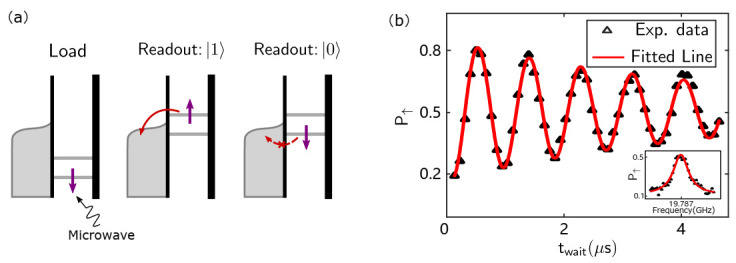
Schematic diagram and measurement result of Rabi oscillation. (**a**) Schematic diagram for Rabi oscillation. (**b**) $P_{↑}$ as a function of $t_{\mathrm{wait}}$. The inset shows $P_{↑}$ as a function of microwave frequency around the resonance frequency ν=19.787 GHz.

**Table 1 nanomaterials-11-02486-t001:** Lever arm $\alpha_{LP}$ for the first four electrons.

Electron Number	Lever Arm $\alpha_{\mathrm{LP}}$ (meV/mV)
1	0.33
2	0.32
3	0.31
4	0.34

## Data Availability

The data presented in this study are available on request from the corresponding authors.

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
