# Peer review of "An Operation Guide of Si-MOS Quantum Dots for Spin Qubits"

_nanomaterials, 2021, doi:10.3390/nano11102486_

Round 1

Reviewer 1 Report

The article by Rui-Zi Hu, Rong-Long Ma and coworkers gives a detailed guide on making and measuring single electron transistors (SETs) which is highly enlightening and well written. There are a few minor typographical errors:

  • Line 3: "long-rang" -> "long-range"
  • Figure 3 caption: "vlotage" -> "voltage"
  • Line 220: "method" -> "methods"

I have no other major reservations about the paper and would support publication of the article with the minor spelling changes as noted.

Reviewer 2 Report

In this article the authors traced a line guide for SI-MOF quantum dots for spin qubits, mainly by showing the basic properties and the commonly method for spin-to-charge conversion. I found the paper pretty complete and it would be a reference for scientific community involved in this topic. I would recommend acceptance after minor revisions:

Line 140: “the forth line decreases” or should be "increases"?

The tendency in Figure 4 are not completely explained or at least correlated with the level arms. Further discussion is needed

From Figure 5d is not clear how t off and t on are extrapolated. Maybe the author want to explain this step.
